# Near-Infrared Fluorescent Sorbitol Probe for Targeted Photothermal Cancer Therapy

**DOI:** 10.3390/cancers11091286

**Published:** 2019-09-01

**Authors:** Sungsu Lee, Jin Seok Jung, Gayoung Jo, Dae Hyeok Yang, Yang Seok Koh, Hoon Hyun

**Affiliations:** 1Department of Otolaryngology-Head and Neck Surgery, Chonnam National University Hwasun Hospital and Medical School, Hwasun 58128, Korea; 2Department of Biomedical Sciences, Chonnam National University Medical School, Hwasun 58128, Korea; 3Institute of Cell and Tissue Engineering, College of Medicine, The Catholic University of Korea, Seoul 06591, Korea; 4Department of Surgery, Chonnam National University Hwasun Hospital and Medical School, Hwasun 58128, Korea

**Keywords:** photothermal therapy, near-infrared fluorescence imaging, tumor targeting, sorbitol, ZW800-1

## Abstract

Photothermal therapy (PTT) using a near-infrared (NIR) heptamethine cyanine fluorophore has emerged as an alternative strategy for targeted cancer therapy. NIR fluorophores showing a high molar extinction coefficient and low fluorescence quantum yield have considerable potential applications in photothermal cancer therapy. In this study, a bifunctional sorbitol–ZW800 conjugate was used as an advanced concept of photothermal therapeutic agents for in vivo cancer imaging and therapy owing to the high tumor targetability of the sorbitol moiety and excellent photothermal property of NIR heptamethine cyanine fluorophore. The sorbitol–ZW800 showed an excellent photothermal effect increased by 58.7 °C after NIR laser irradiation (1.1 W/cm^2^) for 5 min. The HT-29 tumors targeted by sorbitol–ZW800 showed a significant decrease in tumor volumes for 7 days after photothermal treatment. Therefore, combining the bifunctional sorbitol–ZW800 conjugate and NIR laser irradiation is an alternative way for targeted cancer therapy, and this approach holds great promise as a safe and highly efficient NIR photothermal agent for future clinical applications.

## 1. Introduction

Photothermal therapy (PTT), based on the principle of light-to-heat conversion, has great potential for effective cancer treatment because of its high tumor ablation efficiency and minimal side effects on normal tissue [1,2,3,4,5]. PTT is a noninvasive therapeutic intervention for specific target tissues when combined with light excitation and laser-induced heating materials. Importantly, near-infrared (NIR) illumination in the wavelength range 650–950 nm is necessary for the use of various photothermal agents for deep tissue penetration and no damage to normal tissue [6,7,8,9,10]. PTT agents with high absorption in the NIR window efficiently convert light energy into heat energy to induce hyperthermia (>48 °C) and promote tumor necrosis and apoptosis [11,12,13].

Although many polymeric and inorganic nanomaterials are developed for use in PTT, most face the challenges of complicated synthetic processes and unsolved biosafety problem [14,15]. Alternatively, NIR heptamethine cyanine fluorophores, such as indocyanine green (ICG) [16], IR780 [17], and IR825 [18], were recently applied in PTT because of the completive relationship between the NIR fluorescence and photothermal conversion efficiency. By combining the strong NIR extinction coefficient, good photostability, and moderate quantum yield, the NIR fluorophores have significant potential application in PTT.

A zwitterionic NIR fluorophore called ZW800-1 has been previously characterized as an excellent NIR fluorescence imaging agent showing ultralow nonspecific tissue background and rapid elimination from the body through renal filtration [19]. Moreover, ZW800-1 has notable optical properties including a high molar extinction coefficient, a relatively higher quantum yield, and good photostability in aqueous solutions, compared to the clinically available NIR fluorophore ICG [20,21,22]. In terms of the optical properties, ZW800-1 could be considered a potential NIR photothermal agent for PTT.

We have recently reported a tumor-targeting NIR fluorescent probe, sorbitol–ZW800 prepared by the conjugation of sorbitol and ZW800-1 shown in Figure 1A, used for targeted tumor imaging of multiple cancer types such as MCF-7, MDA-MB-231, NCI-H460, and HT-29 [23]. The tumor binding mechanism of the sorbitol–ZW800 conjugate remains under investigation, due to the fact that no cellular uptake was observed in vitro. A preliminary study on the tumor targetability of sorbitol–ZW800 showed excellent in vivo performance in tumor xenograft mouse models.

In this study, we used the sorbitol–ZW800 conjugate for targeted tumor imaging and photothermal treatment in vivo. sorbitol–ZW800 with high tumor targetability showed strong NIR fluorescence in tumor tissue and high photothermal efficacy under NIR laser irradiation, compared to a control group in vivo. Therefore, we demonstrated the bifunctional sorbitol–ZW800 could be a safe and effective theranostic agent for image-guided cancer therapy in clinical application.

## 2. Materials and Methods

### 2.1. Materials

All chemicals and solvents were commercially obtained from Sigma-Aldrich (St. Louis, MO, USA) with the highest purity level and used as received without further purification. ZW800-1 NHS ester was synthesized as described previously [19,20,21].

### 2.2. Conjugation of Amino-Sorbitol to ZW800-1 NIR Fluorophore (sorbitol–ZW800)

Amino-sorbitol (1-Amino-1-deoxy-D-sorbitol; 1.5 µmol, 0.27 mg) was conjugated to ZW800-1 NHS ester (1 µmol, 1 mg) in the presence of *N,N*-diisopropylethylamine (DIEA used for adjusting pH 10; 5 µmol, 0.65 mg) in DMSO (2 mL) at room temperature for 1 h. The reaction mixture was separated using a preparative HPLC system equipped with a PrepLC 150 mL fluid handling unit, a manual injector (Rheodyne 7725i, Waters, Milford, MA, USA), and a 2487 dual wavelength absorbance detector (Waters). Molecular weight of the purified sample was measured by mass spectroscopy using an ultra-performance liquid chromatography (UPLC, Waters) device equipped with micrOTOF-Q II (Bruker, Germany).

### 2.3. Optical Property Analysis

All optical properties were measured in phosphate-buffered saline (PBS), pH 7.4. Absorbance and fluorescence emission spectra of sorbitol–ZW800 conjugate were acquired on a fiber optic Flame absorbance and fluorescence (200–1025 nm) spectrometer (Ocean Optics, Dunedin, FL, USA). The molar extinction coefficient was calculated using the Beer–Lambert law equation. To determine fluorescence quantum yield, ICG dissolved in DMSO (quantum yield = 13%) [19,20] was used as a calibration standard under the condition of matched absorbance at 770 nm. NIR excitation light source was provided by 5 mW of 655 nm red laser pointer (Opcom Inc., Xiamen, China) coupled through a 400 µm core diameter, NA 0.22 fiber (Ocean Optics).

### 2.4. HT-29 Xenograft Mouse Model

Animal study was performed in accordance with protocols approved by the Chonnam National University Animal Research Committee (CNU IACUC-H-2017-64). Male NCRNU nude mice (6 weeks old, ~25 g) were purchased from Orient (Seongnam, Korea). HT-29 cancer cell line was obtained from the American Type Culture Collection (ATCC^®^ HTB-38™). Cancer cells (1 × 10^6^ cells per mouse) were harvested, counted, and suspended in 100 μL of PBS. Then, HT-29 cancer cells were implanted subcutaneously into the right flank of each mouse. When tumors attained a size of 1 cm in diameter, animals were injected intravenously with the sorbitol–ZW800 conjugate and imaged over a certain period of time.

### 2.5. In Vivo Tumor Imaging

In vivo NIR fluorescence imaging was performed using a fluorescence imaging system (FOBI, NeoScience, Suwon, Korea). Fluorescence intensities accumulated in tumors were analyzed using ImageJ version 1.45q (National Institutes of Health, Bethesda, MD, USA). To confirm the in vivo antitumor effect, the macroscopic appearances on each group were observed at determined time intervals for a week. The tumor volumes were measured by the following formula: V = 0.5 × longest diameter × (shortest diameter)^2^.

### 2.6. In Vivo Photothermal Effect Assessment

HT-29 tumor mice were intravenously injected with PBS and sorbitol–ZW800, respectively. At 2 h post-injection, the tumor areas were locally irradiated with laser (1.1 W/cm^2^, *λ* = 808 nm) for 5 min. Temperature changes in the tumor areas were observed by using a FLIR^®^ thermal imager (FLIR Systems, Wilsonville, OR, USA). Data record started from the beginning of the laser irradiation at a step-size of 1 min during the whole period of laser irradiation. At 24 h post-irradiation, tumors were resected from the treated mice for histological examination using hematoxylin and eosin (H & E) staining.

### 2.7. Statistical Analyses

Statistical analyses was performed using a one-way ANOVA followed by Tukey’s multiple comparisons test. Differences were considered to be statistically significant at a level of *p* < 0.05. Data were presented as mean ± S.D. and curve fitting was carried out using Prism version 4.0a software (GraphPad, San Diego, CA, USA).

### 2.8. Histological Analysis

Resected tumors were preserved for H & E staining and microscopic examination. Tumor tissues were fixed in 2% paraformaldehyde and flash frozen in optimal cutting temperature (OCT) compound using liquid nitrogen. Frozen tumor tissues were cryosectioned (10 µm in thickness per slide), stained with H & E, and observed by microscopy. Histological imaging was conducted on a Nikon Eclipse Ti-U inverted microscope system (Nikon, Seoul, Korea). Image acquisition and analysis were carried out using NIS-Elements Basic Research software (Nikon).

## 3. Results and Discussion

### 3.1. Preparation and Characterization of sorbitol–ZW800 Conjugate

To create tumor-targeted sorbitol–ZW800, amino-sorbitol was covalently conjugated to ZW800-1 NHS ester through amide bond formation by a condensation reaction occurred in base condition (Figure 1A). The sorbitol–ZW800 conjugate was separated by a preparative HPLC system with high yield (91%) and purity (94%). The tumor targetability of sorbitol–ZW800 was previously proven in multiple cancer types such as MCF-7, MDA-MB-231, NCI-H460, and HT-29 [23]. In Figure 1B, the mass spectrum showed exact mass of sorbitol–ZW800, indicating successful conjugation and suitable conditions for in vitro and in vivo studies. Since the sorbitol–ZW800 conjugate was highly soluble in aqueous solution, the molar extinction coefficient and quantum yield (*ε* = 246,000 M^−1^cm^−1^, *Φ* = 13.5% in PBS) are relatively higher than ICG (*ε* = 121,000 M^−1^cm^−1^, *Φ* = 9.3% in serum) [19].

Importantly, optical spectra of sorbitol–ZW800 in PBS exhibited the highest absorption at 770 nm and NIR fluorescence peak at 788 nm, respectively (Figure 1C). The absorption peak of sorbitol–ZW800 was mismatched with a 808 nm laser diode used for PTT, leading to the photothermal conversion efficiency being less than that 770 nm laser diode.

### 3.2. Assessment of In Vitro Photothermal Effect

For photothermal imaging, a FLIR^®^ thermal imager equipped with a laser diode (808 nm, 0–2 W) was used to determine the photothermal conversion efficiency of sorbitol–ZW800 conjugate. The 808 nm NIR light, forming laser beam bundle, homogeneously illuminates the target site for a certain period of time, then temperature change was monitored simultaneously. The fixed position of the laser illuminator is important for in vitro and in vivo photothermal imaging, because the laser power density should be precisely measured to show reliable and comparable photothermal images.

The photothermal imaging in vitro was investigated by monitoring the solutions of sorbitol–ZW800 (10 μg/100 μL in PBS; 100 μM) and PBS alone (100 μL) irradiated by a 808 nm NIR laser (1.1 W/cm^2^) for 1 min. The concentration of sorbitol–ZW800 was equivalent to 0.4 mg/kg as a single dose typically used for in vivo studies. As the irradiation time of sorbitol–ZW800 solution increased, the color of the photothermal images rapidly changed from dark blue (corresponding to low temperature) to bright yellow (corresponding to high temperature) within 1 min, compared to the solution of PBS alone (Figure 2). The temperature was elevated from room temperature (25.7 °C) to 86.8 °C for the sorbitol–ZW800 solution, and little changed for PBS solution after 1 min irradiation. This result indicates that the sorbitol–ZW800 solution could absorb the NIR light and convert to a large amount of thermal energy in a time-dependent manner. Upon irradiation by the 808 nm laser with the power density of 1.1 W/cm^2^ for 1 min, however, the fluorescence intensity of sorbitol–ZW800 significantly diminished.

Based on the equation reported by Liu et al. [15], the photothermal conversion efficiency (*η*) of the sorbitol–ZW800 conjugate was calculated as follows:
$$\eta =\frac{hA\Delta T_{\max}-Q_{s}}{I(1-10^{-A\lambda })}$$where *h* is the heat transfer coefficient, *A* is the surface area of the container, ΔT_max_ is the temperature change of the sample solution at the maximum temperature, *I* is the laser power density, *A*_λ_ is the absorbance of sample at 808 nm, and Q_s_ is the heat associated with the light absorbance of the solvent. Following this equation, the photothermal conversion efficiency (*η*) of the sorbitol–ZW800 conjugate was calculated to be 32.6%. This value is relatively higher than that of inorganic photothermal agents such as gold nanoparticles (13–21%) and Graphene oxide (25%) [13].

### 3.3. In Vivo NIR Fluorescence Imaging for Tumor Targetability

By using the single dose (10 nmol, 0.4 mg/kg) optimized previously [23], the real-time tumor accumulation in HT-29 tumor-bearing mice was imaged at different time points after the intravenous injection of the sorbitol–ZW800 conjugate as shown in Figure 3. The fluorescence signals concentrated on the tumor tissue within 1 h, then the fluorescence intensity continuously increased and reached maximum at 2 h. After that time, the tumor fluorescence gradually became weaker due to the metabolism (Figure 3A,B). This result indicates that the sorbitol–ZW800 could be useful for effective tumor-specific imaging, and provide fluorescence-guidance and optimal timing for the photothermal treatment.

We have previously studied the biodistribution of the sorbitol–ZW800 conjugate by imaging the fluorescence in the tumor and major organs resected from the mice. The sorbitol–ZW800 was mainly distributed in the tumor and kidneys with high fluorescence for 24 h [23]. The higher fluorescence in the kidneys suggested that sorbitol–ZW800 were mostly eliminated by renal excretion from the body. The in vivo performance of the sorbitol–ZW800 conjugate may be affected by that of the ZW800-1 fluorophore, except for tumor targetability, because ZW800-1 itself showed no tumor targeting effect until 4 h post-injection (Figure 3C).

### 3.4. Assessment of In Vivo Photothermal Effect

Based on the photothermal effect of sorbitol–ZW800 in vitro, the PTT capability of sorbitol–ZW800 in vivo was subsequently studied on the HT-29 tumor-bearing mouse model under the photothermal imaging system as shown in Figure 4A. Mice were intravenously injected with sorbitol–ZW800 (10 nmol, 0.4 mg/kg) at 2 h prior to laser irradiation and subjected to 808 nm laser (1.1 W/cm^2^) exposure for 5 min. A FLIR^®^ thermal imager was taken to monitor the local temperature variation. The tumor temperature rapidly reached ~55 °C within 2 min after laser irradiation, and moreover the photothermal treatment was maintained until 5 min for effective tumor ablation (Figure 4B). After laser irradiation, only a moderate increase in temperature to ~42.5 °C on the tumor of PBS-injected mice was measured, while a rapid increase in tumor temperature to ~58.7 °C was significantly observed in the group of mice treated with sorbitol–ZW800 (Figure 4C). These results demonstrate that the tumor temperature change was induced by the strong photothermal effect of sorbitol–ZW800.

### 3.5. In Vivo PTT Efficacy

To further confirm the therapeutic effect of the sorbitol–ZW800 conjugate, HT-29 tumor-bearing mice were continuously monitored for 7 days after photothermal treatment (Figure 5A). The tumor sizes in PBS- or sorbitol–ZW800-injected groups were measured every other day. Tumor volumes in the sorbitol–ZW800-treated group rapidly decreased within 4 days after laser irradiation, with only black scars remaining at the original tumor sites. In contrast, tumors in the PBS-injected group increased significantly over time without affecting the laser irradiation (Figure 5B). The results indicate that a combination of sorbitol–ZW800 injection and laser irradiation could inhibit tumor growth. Furthermore, no tumor regrowth was observed in the sorbitol–ZW800-treated group within 10 days after photothermal treatment. The behaviors of sorbitol–ZW800-treated mice after laser irradiation were also monitored during the course of therapy. The treatment group showed no visible signs of physiological weight loss and toxicity within a week. Therefore, we demonstrated that the sorbitol–ZW800 injection with laser irradiation could completely ablate tumors on mice.

Furthermore, the tumors were harvested at 24 h after laser irradiation, and histological analysis was performed using H & E staining in each group (Figure 6). As expected, obvious evidence of cell damage, such as cell shrinkage and nuclear damage, was observed in sorbitol–ZW800-treated tumors exposed to laser irradiation, while the tumor tissue in the PBS and laser-treated group showed vigorous cell proliferation, a tight arrangement, and intact shape without photothermal effect. This result was consistent with the therapeutic efficacy in vivo, and indicates that the sorbitol–ZW800 conjugate could be used for effective tumor phototherapy.

## 4. Conclusions

In this study, we successfully used the sorbitol–ZW800 conjugate as a targeted photothermal agent for cancer therapy, which shows significant photothermal efficiency due to the high tumor targetability of the sorbitol moiety and excellent photothermal property of ZW800-1 heptamethine cyanine fluorophore. The HT-29 tumor mice treated with sorbitol–ZW800 under NIR laser irradiation showed a significant decrease in tumor volumes over 7 days of photothermal therapy. To the best of our knowledge, no previous studies have reported the ZW800-1 NIR fluorophore as a highly efficient NIR photothermal agent for targeted cancer therapy. Therefore, this study highlights the promise of sorbitol–ZW800 as a safe and efficient platform technology for the next generation of in vivo photothermal therapeutic agents.

## Figures and Tables

**Figure 1 cancers-11-01286-f001:**
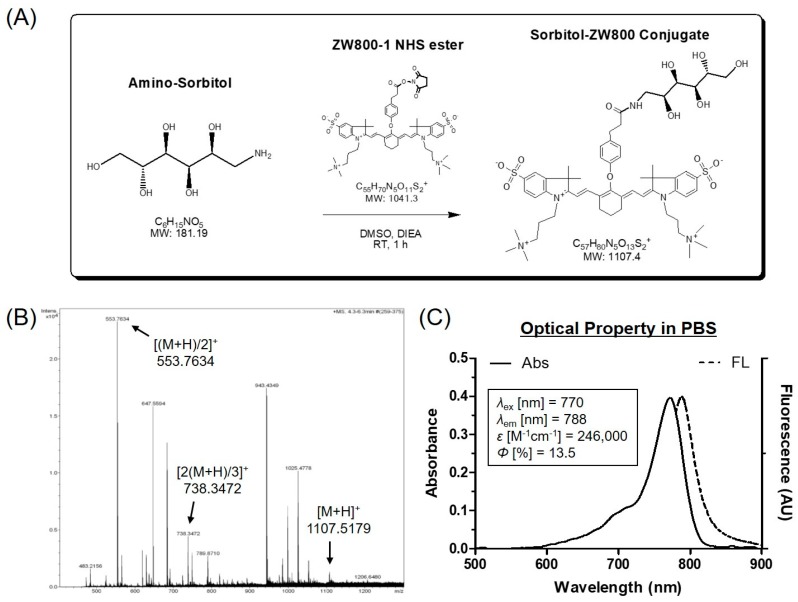
(**A**) Synthetic scheme, (**B**) mass spectrum, and (**C**) optical property of sorbitol–ZW800 conjugate. All optical measurements were performed in PBS, pH 7.4.

**Figure 2 cancers-11-01286-f002:**
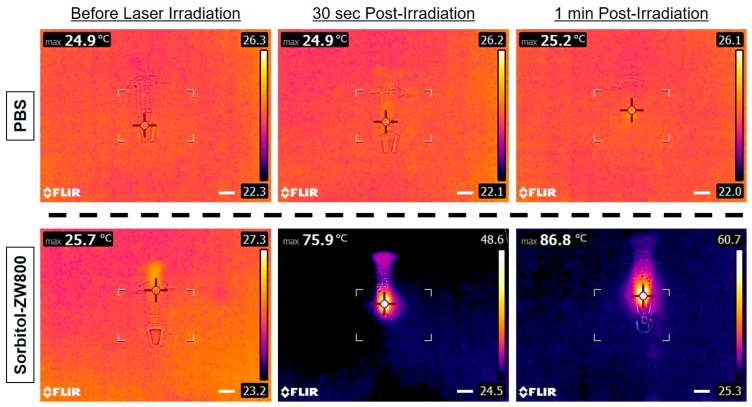
In vitro photothermal images of the sorbitol–ZW800 solution (10 μg/100 μL in PBS; 100 μM concentration is equivalent to 0.4 mg/kg as a single dose) and PBS alone (100 μL) exposed to the 808 nm laser (1.1 W/cm^2^) for 1 min. The maximum temperature was automatically recorded with an infrared thermal camera as a function of irradiation time. Scale bars = 1 cm.

**Figure 3 cancers-11-01286-f003:**
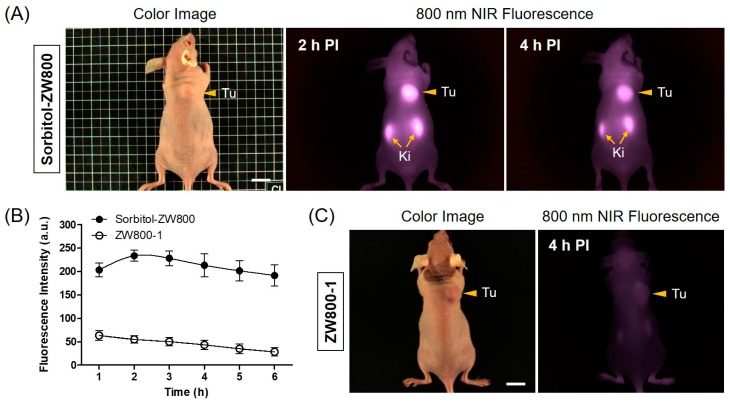
In vivo HT-29 tumor targeting efficiency of sorbitol–ZW800 and ZW800-1 alone. (**A**) NIR fluorescence imaging at 2 h and 4 h post-injection of sorbitol–ZW800. (**B**) Time-dependent fluorescence intensities of tumor sites targeted by sorbitol–ZW800 and ZW800-1 alone. (**C**) NIR fluorescence imaging at 4 h post-injection of ZW800-1 alone. Tumor mice were intravenously injected with sorbitol–ZW800 or ZW800-1 alone (10 nmol, 0.4 mg/kg) and imaged for 24 h. The tumor site is indicated by arrowheads. Abbreviations: Tu, tumor; Ki, kidneys; and PI, post-injection. Scale bars = 1 cm. Images are representative of three independent experiments. All NIR fluorescence images have identical exposure and normalizations. Data are expressed as mean ± S.D. of three independent experiments.

**Figure 4 cancers-11-01286-f004:**
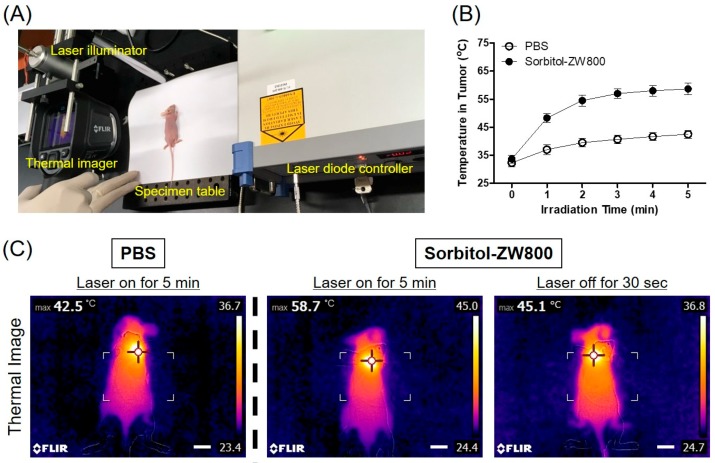
(**A**) A photograph of the photothermal imaging system equipped with a 808 nm laser illuminator, a laser diode controller, and a thermal imager. The 1.1 W/cm^2^ laser power density was determined by laser parameters of 5 mm beam diameter and 0.22 W power. (**B**) Temperature changes at the tumor sites in each treatment group were monitored for 5 min of the 808 nm laser irradiation (1.1 W/cm^2^). Data are expressed as mean ± S.D. of the three independent experiments. (**C**) The whole-body photothermal images of tumor mice at 2 h post-injections of PBS and sorbitol–ZW800 (10 nmol, 0.4 mg/kg), respectively, under exposure to the 808 nm laser irradiation (1.1 W/cm^2^) for 5 min. The maximum tumor temperatures were automatically recorded with an infrared thermal camera as a function of irradiation time. Scale bars = 1 cm.

**Figure 5 cancers-11-01286-f005:**
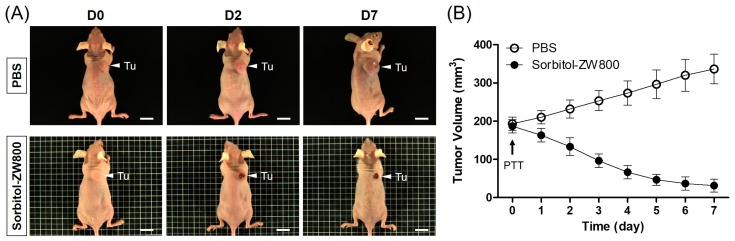
(**A**) In vivo NIR photothermal therapeutic efficacy. Representative photos of tumor size changes treated with 2 h post-injections of PBS and sorbitol–ZW800 (10 nmol, 0.4 mg/kg), respectively, followed by 808 nm laser irradiation (1.1 W/cm^2^) for 5 min. (**B**) Tumor growth rates of each treatment group were monitored for 7 days. Data are expressed as mean ± S.D. of the three independent experiments. Abbreviation: Tu, tumor; and PTT, photothermal therapy. Scale bars = 1 cm.

**Figure 6 cancers-11-01286-f006:**
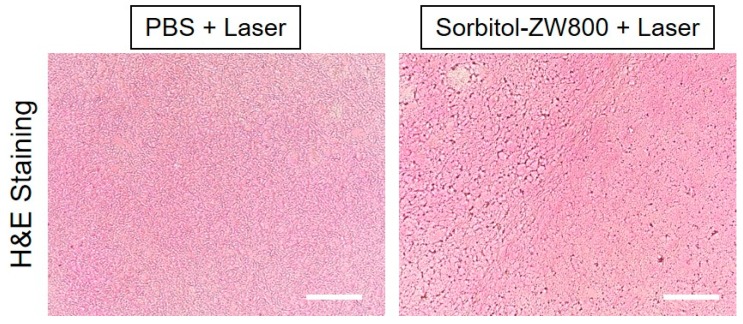
Tumor H & E-stained slices of PBS and sorbitol–ZW800 injected mice at 24 h after laser irradiation. Scale bars = 100 μm.

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
