# Peer review of "Near-Infrared Fluorescent Sorbitol Probe for Targeted Photothermal Cancer Therapy"

_cancers, 2019, doi:10.3390/cancers11091286_

Round 1

Reviewer 1 Report

In this manuscript, Sorbitol-ZW800 conjugate was successfully used as a targeted phoththermal agent for cancer therapy due to the high tumor targetability of sorbitol moiety and excellent photothermal property. Overall, this study is interesting and acceptance is recommended after revision.

What’s the amount of conjugates accumulated in the tumor? Could the authors provide this data? According to the results, Sorbitol has specific tumor targeting performance, could authors provide some basic characterization (e.g., confocal image) about this? Some detailed discussion is also required.

Author Response

In this manuscript, Sorbitol-ZW800 conjugate was successfully used as a targeted photothermal agent for cancer therapy due to the high tumor targetability of sorbitol moiety and excellent photothermal property. Overall, this study is interesting and acceptance is recommended after revision.

What’s the amount of conjugates accumulated in the tumor? Could the authors provide this data? According to the results, Sorbitol has specific tumor targeting performance, could authors provide some basic characterization (e.g., confocal image) about this? Some detailed discussion is also required.

Thank you very much for your kind comments. As the major limitation of optical imaging, the fluorescence imaging using fluorophores is basically difficult to quantify the exact amount of conjugates accumulated in the tumor, unlike nuclear medicine imaging using radioisotopes. Regarding the tumor targetability of Sorbitol conjugate, we have previously investigated the in vitro cell binding assay for MCF-7, MDA-MB-231, NCI-H460, and HT-29 cancer cells, unfortunately, no cell uptake was observed even though the Sorbitol conjugate showed high tumor accumulation in vivo. Although the tumor binding mechanism of Sorbitol conjugate is still unknown, we are trying to identify the tumor targeting mechanism for the future study. We thank the Reviewer for kind understanding on this matter.

We added the explanation in Introduction part as “The tumor binding mechanism of Sorbitol-ZW800 conjugate is still under investigation, because no cellular uptake was observed in vitro”.

Reviewer 2 Report

This study introduces an advanced concept of photothermal therapeutic agents for in vivo cancer imaging and photothermal therapy, which consisted of sorbitol moiety with high tumor targetability and NIR heptamethine cyanine fluorophore with excellent photothermal property. The agent was synthesized through condensation reaction between both the sorbitol and ZW800-1, demonstrating the excellent anti-tumor activity via its photothermal effect by NIR light irradiation. The combination of bifunctional Sorbitol-ZW800 conjugate and NIR laser irradiation is an alternative way for targeted cancer therapy, and this approach holds great promise as a safe and highly efficient NIR photothermal agent for future clinical applications. To be highlighted in this journal, some issues were should be clarified, as stated belows:

Figure 1 shows the information such as chemical structure and optical property of Sorbitol-ZW800 conjugate. This manuscript said the material was synthesized by condensation reaction. I think the authors need to address the mechanism and more detailed description regarding the synthesis. This manuscript said that Sorbitol-ZW800 conjugate was synthesized by condensation under base condition. But, there is no information about pH condition in section 2.2. Please add more detail information of synthesis condition. In this manuscript, all optical measurements were performed at 37 oC in 100 % FBS solution, while heat generation test by NIR light irradiation and animal test were conducted with PBS solution. I think it is unusual to use FBS for optical measurements, and I am wondering why the authors choose FBS to show the optical property of Sorbitol-ZW800 conjugate. Figure 2 represented the in vitro photos during 1 min of NIR irradiation to demonstrate the photothermal ability of the system. To more clarify the ability, the authors should provide the information of the exact time for each photo. Or they can In section 3.5, please revise ‘repidly’ in 4th line, to ‘rapidly’.

Author Response

This study introduces an advanced concept of photothermal therapeutic agents for in vivo cancer imaging and photothermal therapy, which consisted of sorbitol moiety with high tumor targetability and NIR heptamethine cyanine fluorophore with excellent photothermal property. The agent was synthesized through condensation reaction between both the sorbitol and ZW800-1, demonstrating the excellent anti-tumor activity via its photothermal effect by NIR light irradiation. The combination of bifunctional Sorbitol-ZW800 conjugate and NIR laser irradiation is an alternative way for targeted cancer therapy, and this approach holds great promise as a safe and highly efficient NIR photothermal agent for future clinical applications. To be highlighted in this journal, some issues were should be clarified, as stated belows:

Figure 1 shows the information such as chemical structure and optical property of Sorbitol-ZW800 conjugate. This manuscript said the material was synthesized by condensation reaction. I think the authors need to address the mechanism and more detailed description regarding the synthesis. This manuscript said that Sorbitol-ZW800 conjugate was synthesized by condensation under base condition. But, there is no information about pH condition in section 2.2. Please add more detail information of synthesis condition.

Thank you very much for your kind comments. As the Reviewer’s suggestion, we newly added the synthetic scheme of Sorbitol-ZW800 conjugate in Figure 1A, and more information about pH condition in section 2.2. described as “Amino-sorbitol (1-Amino-1-deoxy-D-sorbitol; 1.5 µmol, 0.27 mg) was conjugated to ZW800-1 NHS ester (1 µmol, 1 mg) in the presence of N,N-diisopropylethylamine (DIEA used for adjusting pH 10; 5 µmol, 0.65 mg) in DMSO (2 mL) at room temperature for 1 h.”.

In this manuscript, all optical measurements were performed at 37 oC in 100 % FBS solution, while heat generation test by NIR light irradiation and animal test were conducted with PBS solution. I think it is unusual to use FBS for optical measurements, and I am wondering why the authors choose FBS to show the optical property of Sorbitol-ZW800 conjugate.

This is an important point. Because blood plasma is a protein-rich solution, we measured the optical properties of Sorbitol-ZW800 conjugate in serum conditions. Responding to the Reviewer’s suggestion, we revised the optical properties of Sorbitol-ZW800 conjugate measured in the PBS solution as shown in Figure 1B. 

Figure 2 represented the in vitro photos during 1 min of NIR irradiation to demonstrate the photothermal ability of the system. To more clarify the ability, the authors should provide the information of the exact time for each photo.

We apologize for the confusion. The in vitro thermal images were measured at 1 min after laser irradiation. For your information, we added more images measured at 30 sec after laser irradiation as shown in Figure 2. 

Or they can In section 3.5, please revise ‘repidly’ in 4th line, to ‘rapidly’.

We apologize for the typo and revised accordingly.

Reviewer 3 Report

There are a few grammatical issues with the introduction and results section, as well as at least one typo in the conclusion. Please correct.

The authors should refer to the chemical structure of ZW800-sorbitol conjugate on the inset of Figure 1 in the introduction. The authors should also report the yield and % purity of the conjugate after purification in the results section. Capitalize “Flame” in the name of the spectrometer (this is the product name).

Please provide details on how the molar absorption coefficient and quantum yield of the conjugate was determined (method, wavelength, solvent, etc.) in the methods section. Also, please clarify if the numbers for these variables provided on page 3 out of 9 are for the conjugate and ICG in serum, as written, or in aqueous solutions in general. This is important as these might be quite different for ICG when bound to serum albumin.

Figure 1 legend then specifies that the molar absorption coefficient and quantum yield provided are those of the fluorophore alone instead. This contradicts the last paragraph of page 3 out of 9.

Please provide an image or schematic of the laser illuminator/thermal imager combo to better understand the positioning of the source and detectors. Also, what is the beam diameter and how was the laser optical power density measured for determination of photothermal conversion efficiency as well as for in vitro and in vivo photothermal measurements?

The following sentence on page 4 is problematic and needs to be removed or supported with appropriate data: “Although high concentrations of Sorbitol-ZW800 enhanced the photostability and photothermal conversion efficiency, the optimal dose of Sorbitol-ZW800 was used for in vivo study to prevent nonspecific binding, uptake and retention.” First, high concentrations do not enhance photostability or photothermal conversion efficiency as these are intrinsic properties of the molecule. In addition, the authors did not investigate neither of these properties. No photostability data was provided, although it should definitely be reported by both temperature measurements over time as well as acquisition of absorption/fluorescence data over time of irradiation. Second, photothermal conversion efficiency is a quantity that describes what percentage of the laser power absorbed is converted into heat. This was not measured by the authors. The authors should measure this property for the conjugate. Finally, the authors must show the temperature profile of the conjugate solutions over the time frame of irradiation used for in vivo studies (5 minutes).

Data of tumor accumulation over time should be plotted for ZW800 alone (nontargeted formulation) on Figure 3B. In addition, if possible provide additional time points past 6 hours to show how rapidly the agent is cleared from the tumor.

Please provide error bars on Figure 4B. Also, if possible, show how quickly the tumor temperature decreases after the 5 minutes of laser irradiation.

Author Response

There are a few grammatical issues with the introduction and results section, as well as at least one typo in the conclusion. Please correct.

We apologize for the typo and grammatical issues. This manuscript has been carefully checked by a professional English editing service.

The authors should refer to the chemical structure of ZW800-sorbitol conjugate on the inset of Figure 1 in the introduction. The authors should also report the yield and % purity of the conjugate after purification in the results section.

This is an important point. We referred to the chemical structure of Sorbitol-ZW800 conjugate in the Introduction part described as “…. a tumor-targeting NIR fluorescent probe, Sorbitol-ZW800 prepared by conjugation of sorbitol and ZW800-1 shown in Figure 1A, used for….”. Also, we added the information about the yield and purity of conjugate in the section 3.1 described as “The Sorbitol-ZW800 conjugate was separated by a preparative HPLC system with high yield (91%) and purity (94%).”. 

Capitalize “Flame” in the name of the spectrometer (this is the product name). 

We revised the product name “Flame” accordingly.

Please provide details on how the molar absorption coefficient and quantum yield of the conjugate was determined (method, wavelength, solvent, etc.) in the methods section. Also, please clarify if the numbers for these variables provided on page 3 out of 9 are for the conjugate and ICG in serum, as written, or in aqueous solutions in general. This is important as these might be quite different for ICG when bound to serum albumin.

This is a very important point. We added the experimental details in the section 2.3 described as below,

“All optical properties were measured in phosphate-buffered saline (PBS), pH 7.4. Absorbance and fluorescence emission spectra of Sorbitol-ZW800 conjugate were measured using fiber optic Flame absorbance and fluorescence (200–1025 nm) spectrometer (Ocean Optics, Dunedin, FL, USA). Molar extinction coefficient was calculated by using the Beer-Lambert Law equation. To determine fluorescence quantum yield, ICG dissolved in DMSO (quantum yield = 13%) [19,20] was used as a calibration standard under the condition of matched absorbance at 770 nm. NIR excitation was provided by 5 mW of 655 nm red laser pointer (Opcom Inc., Xiamen, China) coupled through a 400 µm core diameter, NA 0.22 fiber (Ocean Optics, Dunedin, FL, USA).”.

As the Reviewer’s comment, the optical properties of ICG are quite variable according to the solvents, especially in aqueous solutions between serum- and serum-free solutions, while ZW800-1 shows similar optical properties in those solutions. Thus, we cited the better optical properties of ICG from the previous study [Ref #19. Synthesis and in vivo fate of zwitterionic near-infrared fluorophores. Angew. Chem. Int. Ed. 2011, 50, 6258–6263.], and compared with that of ZW800-1 conjugate.

Figure 1 legend then specifies that the molar absorption coefficient and quantum yield provided are those of the fluorophore alone instead. This contradicts the last paragraph of page 3 out of 9.

We apologize for the confusion. We investigated the optical properties of Sorbitol-ZW800 conjugate measured in the PBS solution as shown in Figure 1B, and revised the Figure 1 legend. 

Please provide an image or schematic of the laser illuminator/thermal imager combo to better understand the positioning of the source and detectors. Also, what is the beam diameter and how was the laser optical power density measured for determination of photothermal conversion efficiency as well as for in vitro and in vivo photothermal measurements?

This is an important point. We added the photograph of photothermal imaging system in Figure 4A and the information of beam diameter and laser power density in Figure 4A legend described as “Figure 4. (A) A photograph of the photothermal imaging system equipped with an 808 nm laser illuminator, a laser diode controller, and a thermal imager. The 1.1 W/cm2 of laser power density was determined by laser parameters of 5 mm beam diameter and 0.22 W power.”. 

The following sentence on page 4 is problematic and needs to be removed or supported with appropriate data: “Although high concentrations of Sorbitol-ZW800 enhanced the photostability and photothermal conversion efficiency, the optimal dose of Sorbitol-ZW800 was used for in vivo study to prevent nonspecific binding, uptake and retention.” First, high concentrations do not enhance photostability or photothermal conversion efficiency as these are intrinsic properties of the molecule. In addition, the authors did not investigate neither of these properties. No photostability data was provided, although it should definitely be reported by both temperature measurements over time as well as acquisition of absorption/fluorescence data over time of irradiation. Second, photothermal conversion efficiency is a quantity that describes what percentage of the laser power absorbed is converted into heat. This was not measured by the authors. The authors should measure this property for the conjugate. Finally, the authors must show the temperature profile of the conjugate solutions over the time frame of irradiation used for in vivo studies (5 minutes).

This is important points. We agree to the Reviewer’s comments and the sentence was removed in the section. The photostability of ZW800-1 has been reported previously as cited in Ref. #22. (The development of a highly photostable and chemically stable zwitterionic near-infrared dye for imaging applications. Chem. Commun. 2015, 51, 3989–3992.). Although ZW800-1 showed relatively higher photostability than ICG, the fluorescence intensity significantly decreased after laser irradiation. Regarding the photothermal conversion efficiency, we calculated the value and described in the section 3.2 as below,

“Based on the equation reported by Liu et al. [15], the photothermal conversion efficiency (η) of Sorbitol-ZW800 conjugate was calculated as follows:

where h is the heat transfer coefficient, A is the surface area of the container, ΔTmax is the temperature change of the sample solution at the maximum temperature, I is the laser power density, Aλ is the absorbance of sample at 808 nm, and Qs is the heat associated with the light absorbance of the solvent. Following this equation, the photothermal conversion efficiency (η) of Sorbitol-ZW800 conjugate was calculated to be 32.6%. This value is relatively higher than inorganic photothermal agents such as gold nanoparticles (13~21%) and Graphene oxide (25%) [13].”.

Also, we added more thermal images of the conjugate solutions measured at 30 sec after laser irradiation as shown in Figure 2. Although we tried to measure the in vitro thermal images for 5 min laser irradiation, the conjugate solution began to boil in the EP-tube over 1 min laser irradiation, then we turned off the laser source for safety. We thank the Reviewer for kind understanding on this matter.

Data of tumor accumulation over time should be plotted for ZW800 alone (nontargeted formulation) on Figure 3B. In addition, if possible provide additional time points past 6 hours to show how rapidly the agent is cleared from the tumor.

As the Reviewer’s suggestion, the time-dependent tumor accumulation for ZW800-1 alone was added in Figure 3B. Previously, we reported that the Sorbitol-ZW800 conjugate was accumulated in tumors over 24 h with high fluorescence intensity, cited as Ref. #23 (Near-infrared fluorescent sorbitol probe for tumor diagnosis in vivo. J. Ind. Eng. Chem. 2018, 64, 80–84.). Since the Sorbitol-ZW800 conjugate is not cleared rapidly from the tumor for 24 h, we investigated the optimal time point for the best photothermal effect within a short period of time.

Please provide error bars on Figure 4B. Also, if possible, show how quickly the tumor temperature decreases after the 5 minutes of laser irradiation.

In response to the Reviewer’s suggestion, we added the error bars on Figure 4B and the thermal image at 30 sec after laser off in Figure 4C.

Round 2

Reviewer 1 Report

My question has been addressed.

Reviewer 3 Report

No additional edits needed